# Glass-Ceramics Processed by Spark Plasma Sintering (SPS) for Optical Applications

**Babu Singarapu [1,2,\*], Dušan Galusek [1,3] , Alicia Durán [2] and María Jesús Pascual [2,\*]**

[1]  FunGlass, Alexander Dubček University of Trenčín, Študentská 2, 911 50 Trenčín, Slovakia; dusan.galusek@tnuni.sk
[2]  Ceramics and Glass Institute (CSIC), C/Kelsen 5, Campus de Cantoblanco, 28049 Madrid, Spain; aduran@icv.csic.es
[3]  Joint Glass Centre of the IIC SAS, TnUAD, and FChFT STU, FunGlass, Alexander Dubček University of Trenčín, 911 50 Trenčín, Slovakia
\*  Correspondence: babu.singarapu@tnuni.sk (B.S.); mpascual@icv.csic.es (M.J.P.)

**Abstract:** This paper presents a review on the preparation of glass-ceramics (GCs) and, in particular, transparent GCs by the advanced processing technique of spark plasma sintering (SPS). SPS is an important approach to obtain from simple to complex nanostructured transparent GCs, full densification in a short time, and highly homogeneous materials for optical applications. The influence of the different processing parameters, such as temperature, pressure, sintering dwell time on the shrinkage rate, and final densification and transparency, are discussed and how this affects the glass material properties. Normally, transparent glass-ceramics are obtained by conventional melt-quenching, followed by thermal treatment. Additionally, the GC scan is produced by sintering and crystallization from glass powders. Hot pressing techniques (HP) in which the source of heating is high-frequency induction can be also applied to enhance this process. In the case of transparent ceramics and glass-ceramics, spark plasma sintering is a promising processing tool. It is possible to enhance the material properties in terms of its compactness, porosities, crystallization, keeping the size of the crystals in the nanometric scale. Moreover, the introduction of a high concentration of active gain media into the host matrix provides functional glass-ceramics systems with enhanced luminescence intensity through reducing non-radiative transitions like multi phonon relaxation (MPR) and cross relaxations (CR), thus providing longer lifetimes. More effort is needed to better understand the sintering mechanisms by SPS in transparent GC systems and optimize their final optical performance.

**Keywords:** spark plasma sintering; glass-ceramics; pressure; density; transparency; optical applications

---

## 1. Introduction

Sintering through conventional heat treatment in a furnace (free or pressure less) is a high energy and time-consuming method to achieve the desired densification and crystallinity in ceramic and glass-ceramic materials. Pressure-assisted techniques are hot pressing (HP) and spark plasma sintering (SPS). The main drawback of HP is the slow heating rate (5–10 °C·min$^{-1}$) and the longer times to heat the compact powder through conduction and external source heating element. Researchers are continuously searching for new innovative sintering technologies. Hence, in this context, an alternative process is SPS [1].

SPS is a material processing method, through the production of spark plasma applying pressure, temperature, and current, simultaneously. This method reduces considerably the sintering temperature,

provides very fast heating rates, very short holding times, and allows highly transparent materials to be obtained [2]. There is a wide literature available on SPS applied to get ceramic materials with desired properties, as well as other materials like metals, alloys, cermets, intermetallic compounds, composites, and amorphous structures.

Some of the advantages of SPS processing [3] related to the supplying of direct electric current through the powder particles are the following:

- It reduces the sintering temperature and time through applying pressure, also inhibiting crystal growth.
- It reduces flaws, voids, and microcracks, and thus favors to improve properties, such as mechanical and optical.
- It enhances the homogeneity of material characteristics, which is very important to get high transmittance magnitudes and more luminescence intensity in the case of optical materials.
- It's possible to retain the initial shape (near net shape) after SPS.

In order to achieve the required microstructure of the sintered body, it is essential to know which variables are involved in the sintering process and how to control them. These variables are mainly divided into two categories: material variables and process parameters [4], as shown in Table 1.

**Table 1.** Variables affecting sinterability and microstructure.

| | |
|---|---|
| Variables related to the starting powders (material variables) | Powder: shape, size, size distribution, agglomeration, mixture, etc. Chemistry: composition, impurities, non-stoichiometry, homogeneity. |
| Parameters related to sintering conditions (process variables) | Temperature, time, pressure, atmosphere, heating and cooling rate, etc. |

SPS has been used for the preparation of different types of structural ceramic materials (carbides, oxides, borides, MAX phase, and silicides) and functional ceramic materials [2,5]. Nanostructured materials show enhanced physical and mechanical properties. Nevertheless, due to interrelations between densification and grain growth, the control of the nanostructure within the bulk material is a difficult task.

The above difficulties have been solved by numerous authors approaching SPS sintering in different ways.

A two-stage sintering process was adopted by Jow-Lay Huang [6] to prepare silicon nitride ($Si_3N_4$) ceramics, obtaining good densification with control of the grain size. Later, Chen and Wang [7] followed a similar two-stage sintering process with a controllable microstructure by studying the effect of several dopants (Mg and Nb) and the initial powder features (particle size, molding processing, and microstructural homogeneity).

The high-content $Al_2O_3$-$Y_2O_3$-doped SPS-sintered silicon nitride was investigated by O.A. Lukianova1 et al. [8]. Fully dense silicon nitride ceramics with the main phase of $\alpha$-$Si_3N_4$ equiaxed hexagonal grained microstructures were obtained. Calculated grain size from 200 to 530 nm, the high elastic modulus of 288 GPa, and high hardness of 2038 HV were observed from spark plasma sintered at 1550 °C. Silicon nitride with elongated $\beta$-$Si_3N_4$ grains, higher hardness of 1800 HV, the density of 3.25 g/cm$^3$, and Young's modulus of 300 GPa at 1650 °C was observed [9].

Nobuyuki Tamari et al. [10] in 1995 prepared silicon carbide materials by SPS and characterized their densification and mechanical properties. Silicon carbide, alumina (additive), and yttria (additive) powders were used as precursors. The powders were also used for hot pressing in order to compare both techniques. The sintering conditions were the same for both—the temperature of 1500–2000 °C, pressure of 30 MPa, and 5 min of holding time. In the case of HP, temperatures below 1800 °C provided densification of 90% or less, but at 2000 °C, densification reached 98% or more. In the case of

SPS, sintering temperatures of 1700, 1750, and 1800 °C allowed densification of 70%, 90%, and 98%, respectively. Sintering temperature then decreased when using SPS. This was due to the self-heat generation between particles through electrical discharge. SiC grain size was found to be similar in both processes. The mechanical fracture toughness of 98% densified SiC sintered by SPS was about 3.4 MPa·m$^{1/2}$, which was about 10% higher than that of the HP technique processed sample.

In 2007, a Sweden scientist Nygren [11] reported a paper on SPS processing of nanostructured oxide ceramics ($Bi_4Ti_3O_{12}$, $BaTiO_3$, and $SrTiO_3$), especially on the grain growth factor versus grain size of the starting precursor powder.

Hwan-Cheol Kim et al. in 2007 [12] prepared tungsten carbide (WC) and tungsten carbide mixed with cobalt (WC–Co) materials by SPS. They prepared dense and hard materials by employing very short holding times (<2 min) at the sintering temperature. The tungsten carbide (WC) samples densified to 97.6% in a time of 110 s and at a temperature of 1600 °C. In the case of tungsten carbide mixed with cobalt (WC–Co) materials, the samples densified to 99.2% in only 65 s at 1150 °C. Cobalt addition caused a significant decrease in the sintering temperature. The authors concluded that the densification mechanism included carbide particle rearrangement and increment of the diffusion due to the viscous flow of the cobalt binder. There was no appreciable variation related to grain growth, the grain size being 380 nm in both materials.

Mats Nygren and Zhijian Shen [3] published in 2013 a book chapter 8 on "Hot Pressing and Spark Plasma Sintering" in A comparative study of both sintering techniques. They reviewed and outlined the different materials, highlighting the applicability of SPS for different materials and potential industrial applications like ceramic armors, transparent ceramic lenses and windows, thermoelectric, bioactive, and functional graded ceramic materials, and ultra-high temperature materials. However, there was no mention of the applicability of SPS processed materials, such as glasses and transparent glass-ceramics.

Since the invention of the glass-ceramics (GCs) first by Stookey [13,14], there was enormous progress on glass-ceramic science and technology. In recent years, many efforts are focused on GCs, especially for optical applications. The control of nucleation and crystallization processes is fundamental for the successful preparation of GCs materials with improved functionalities. The efficiency of GCs depends on the various factors, such as size, shape, surface morphology and physical properties (density and transparency), and precipitation of desired crystalline phases. GCs are normally produced by melt-quenching, followed by suitable thermal treatment to promote nucleation and crystallization of the interested crystalline phases. The preparation of GCs through powder technology implies the combination of viscous flow sintering together with crystallization. Normally, sintering must be completed before the beginning of crystallization for obtaining perfectly dense materials. In the last years, novel sintering techniques, and particularly SPS, have been used to obtain well-densified transparent glass-ceramic materials [15–17].

Single crystals, ceramics, glasses, and glass-ceramics (GCs) are important materials in optical applications with the condition of high transparency, which allows light transmission without scattering. Transparent ceramics synthesized by SPS technique, such as $ZrO_2$ [18], $Al_2O_3$ [19,20], MgO [21], non-oxide AlN [22], yttrium aluminum garnet (YAG) [23,24], and $MgAl_2O_4$ [25,26] have been widely reported, even though these materials appear as transparent in nature, and the same phenomenon does not apply for the transparent glass-ceramics.

The physical properties of transparent GCs are influenced by crystallization kinetics, crystallinity, microstructure, the nature of the glassy and crystalline phases, and their chemical composition. The optical properties, transparency, in particular, can be controlled by the precipitation of nanocrystals, similar refractive index values for the glass matrix and the crystalline phase, and low birefringence of the crystalline phase [27,28]. Transparent GCs are very important materials for optical applications because of the possibility of mass production, shaping as required, and high incorporation of dopant concentration. In particular, oxyfluoride transparent glass-ceramics in which fluoride nanocrystals can precipitate offer a high solubility for the rare-earth ions in the crystal phase, providing more

efficient optical emissions [29,30]. Transparent GCs can fulfill the gap between glasses and ceramics, in particular, as optical materials, and SPS turns to be an alternative method to produce dense materials.

It is noteworthy that, to the best of the author's knowledge, no review article about state-of-the-art is available that summarizes the benefits by SPS towards the development of transparent GCs. Hence, the major motivation for this review is to shade light in this area. After short information about the SPS method (brief history and sintering mechanism), this review paper has focused on the SPS processing of transparent ceramics and transparent glass-ceramic materials for optical applications and the comparison with other processing methods, evaluation of the mechanisms of sintering and crystallization, and final properties. Some recent studies concerning GCs prepared by SPS have been included and discussed. However, there is no relevant literature about transparent glass-ceramics, the effect of glass powder size and their distribution, sintering behavior, and the effect of processing parameters on the transparency. Finally, some future prospects about SPS processing are given.

## 2. History of Spark Plasma Sintering

SPS also called pressure-assisted pulse energizing process, pulsed electric current sintering (PECS), electric current-assisted/activated sintering (ECAS), field-assisted sintering technology (FAST), or plasma-activated sintering (PAS) is a promising technology of the 21st century for the fabrication of novel materials [31].

Since the last 20 years, the SPS method has reached pronounced importance in the powder metallurgy industry and in powder technology in general for the preparation of advanced materials. As mentioned in the introduction, it is well-known that the SPS is an advanced technique to obtain homogeneous, highly dense nanostructured sintered compacts, fine ceramics, composite materials, new wear-resistant materials, thermoelectric semiconductors, and biomaterials. An important number of SPS technology-based products are available in the market from Japanese industries. The SPS is now touching various action points through the scientific academia -R&D departments and engineered materials-practical industry [2].

The SPS system was first introduced in Germany around 1910 as an electric source oriented to densify materials. In the USA, G.F. Tayler patented the first resistance sintering method for sheet metals in 1933 [32]. Thereafter, G.D. Cremer obtained US patent for the method of sintering copper, brass, or aluminum powder materials [33]. They were considered as the origin of a current hot pressing (HP) technique that commonly applies a high-frequency induction heating method. Inoue et al. developed SPS based on the idea of using plasma in an electric discharge machine for sintering metals and ceramics in the early 1960s. Dr. Kiyoshi Inoue of Japax Inc. originally invented SPS in Japan as "spark sintering (SS)" in 1962 and got patent rights in 1966 [34,35]. Finally, Sumitomo Coal Mining Co., Ltd. (Tokyo, Japan) introduced the present SPS in 1989.

Figure 1 shows a typical SPS system, which consists of a hydraulic press with a (mostly) vertical single pressurization axis. The pressure is transferred via two steel cylinders (rams); there are two graphite spacers and two graphite punches between upper and lower punch electrodes. The sintered powder is stacked in a cylindrical die and pressed between punches. Everything is built in a water-cooled vacuum chamber. The water-cooled electrodes are connected to an electric power supply. The power supply produces electric current flowing through graphite punches, sintered powder, and particularly through a graphite die. Therefore, the die and the punches are made of an electrically conductive material, which must be able to resist high temperature and pressure. The powder is located in the middle of the die [31,36].

In 2009, Salvatore Grasso et al. [37] published a paper on "Electric current-activated/assisted sintering (ECAS): a review of patents 1906–2008". In this report, they concluded, "In the past century, the simultaneous development of basic ECAS apparatuses and peripheral units was fundamental in overcoming intrinsic technological limitations and in optimizing ECAS processes with respect to (a) product size, (b) microstructure homogeneity, particularly for large compacts, (c) process reproducibility

and (d) processing parameters". But, still, to date, these fundamental problems decide the applicability of their use in industry.

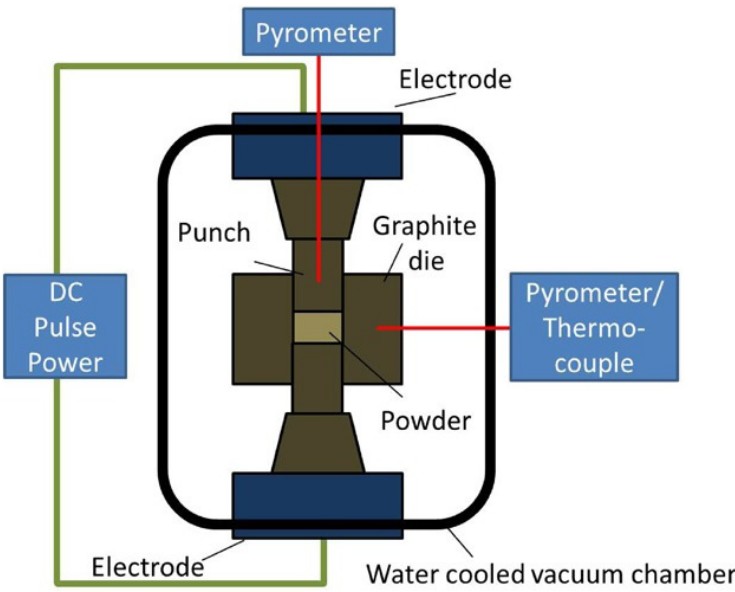

**Figure 1.** Schematic of spark plasma sintering unit. Reprinted with permission from Advanced Engineering Materials [31].

## 3. Mechanism of Spark Plasma Sintering

SPS is a more advanced and efficient sintering technique in terms of its operating parameters compared with the pressureless sintering (PLS), hot press (HP) sintering, and hot isostatic pressing (HIP). SPS makes possible sintering at lower temperatures and in a short period of time by charging the powder particles with electrical energy and effectively applying a high-temperature spark plasma generated at an initial stage of pulse energizing particles in short sintering time. Pulse current energizing provides enhancement of sinterability and densification rate on the material. An electromagnetic field and/or joule heating is applied by continuous ON/OFF DC pulsed high electric current with low voltage. The technique appears like HP heated by a radiative furnace. However, the SPS treated samples can reach relevant qualities like structurally tailoring effect, grain growth control, enhancement of electro-migration, and strong preferential orientation effect [2,8].

The SPS process is a dynamical non-equilibrium processing phenomenon related to the passage of plasma from initial to final stages via reacted material characteristics. However, the ON/OFF DC pulse energizing method is one of the best-implemented mechanisms of SPS processing and thus creates (1) spark plasma, (2) spark impact pressure, (3) Joule heating, and (4) an electrical field diffusion effect. In the initial stage of the SPS presented in Figure 2, DC pulsed voltage (ON mode) does apply through die and punches made of a graphite material, powder material, and the subsequent heating (Joule heating) and densification of the powder. An electric noise is heard during the process, attributed to the plasma generation. At the final stage, the formation of the spark discharges is located in the gaps between particle surfaces. At the time of ON mode, the powder particle surfaces are more purified, producing materials with surface-layer composition and microstructure different from that of the core. In the OFF mode, plasma passes throughout the volume of the sintered powder, followed by vaporization, melting of surface, and neck formation. The large pulse energy generates an electromagnetic field effect, such as an electro-migration and preferential orientation of the crystalline phase [8,38].

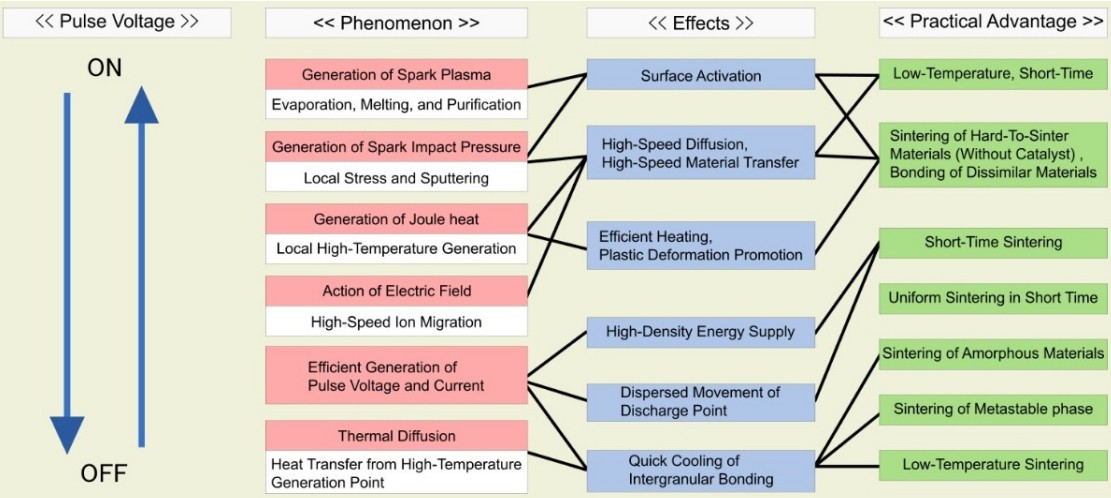

**Figure 2.** Effects of ON-OFF DC pulse energizing. Reprinted with permission from the InTechopen publisher [38].

The sintering mechanism of nano grained (NG-YAG) ceramics at high pressure was proposed by Wang et al. [39]. According to these authors, the sintering kinetic process can be divided into three stages. In the first step, the sintering pressure promotes a plastic deformation period. At the middle stage, the grain boundary viscous sliding controls the process. The final step, under high pressure, includes the transformation from grain boundary viscous sliding to grain boundary diffusion.

In the case of GCs, a viscous flow sintering mechanism occurs when the volume fraction of liquid is sufficiently high so that full densification of the compact can be achieved by viscous flow of grain-liquid mixture without having any grain shape change during densification. The viscous flow mechanism, which was first proposed by Frenkel [40,41], can be applied in the sintering of viscous materials like glass, which follows the behavior of a Newtonian fluid. Weinberg [42] suggested that the application of high pressure increases glass viscosity and reduces the crystal growth rate. Crystallite size decrease corresponding to a pressure increase suggests a suppressed or hindered crystal growth. The application of high pressure is thought to increase the viscosity of the glass surrounding the pre-nucleated crystals, reducing the atomic movement and diffusion across the crystal–liquid interface [43].

From now on, some examples of transparent ceramics and glass-ceramics processed by SPS are discussed.

## 4. Transparent Ceramics

$ZrO_2$ possesses peculiar mechanical and functional properties, especially yttria-stabilized zirconia (YSZ), which has been widely investigated. It has been prepared by glycine-nitrate process and ball milling and the nanocrystalline precursor powders of 8 mol% YSZ ceramics used for SPS. The specimens sintered at 1200 and 1250 °C were opaque due to high porosity that causes the lowering of transmittance. Yttria-stabilized zirconia transparent ceramics were obtained at 1300 °C by SPS route and within a short period of time (5 min). With the rise in temperature, direct transmittance decreased due to scattering by grain boundaries and a higher number of oxygen vacancies [44].

Transparent alumina/ceria ceramics were also obtained by SPS [45]. It was found that transparency was enhanced upon the addition of ceria nanoparticles in transparent alumina material. At the initial stage of sintering, ceria nanoparticles contribute to densification, while, at the later stage, they delay alumina grain growth. The best sintering conditions were fixed at 1430 °C, 80 MPa, and 2 min.

SPS transparent YAG ceramics [46] were fabricated at 1400 °C, the pressure of 50 MPa, and 3 min time, using starting commercial nanocrystalline YAG precursor powders. An increase in the SPS duration and pressure significantly increased the density, especially at temperatures below 1400 °C. The observed microstructure was in agreement with densification by nano-grain rotation and sliding

at lower densities, followed by curvature-driven grain boundary migration and normal grain growth at higher densities. Residual nanosize pores at the grain boundary junctions were an inherent microstructure feature due to the SPS process. The direct transmission intensity was lower at the lower wavelengths, likely due to the higher Rayleigh scattering in this range, the scattered intensity being inversely proportional to the fourth power of the wavelength.

$Dy^{3+}$ (3 at%)-doped $Y_2O_3$ ceramics were also fabricated by the SPS process [47]. The starting precursor powders were weighed and sieved through 60 mesh screen and used for further SPS measurements under vacuum. The sintering conditions, 600 °C for 5 min, further increased to 1100 °C for 3 min and finally to 1400 °C for 30 min. Similarly, four samples at different final temperatures like 1450, 1500, 1550, and 1600 °C were also prepared. From XRD profiles, it was observed that no significant changes occurred with the addition of dopant in the ceramic $Y_2O_3$ host matrix. A large content of pores in the grain boundary was observed in the sample prepared at 1400 °C. When increasing the sintering temperature to 1550 °C, there was a decrease in the pores' size and increase in the grain size together with a uniform distribution of grains. At 1600 °C, an abnormal increase in grain growth took place, affecting optical transmittance. The maximum direct transmittance (74.5%, at 574 nm) was obtained for the sample sintered at 1550 °C. The laser parameters were assessed from applying the Judd–Ofelt theory. The prepared transparent $Y_2O_3$:Dy ceramic materials were found useful for yellow laser emission corresponding to $^4F_{9/2} \rightarrow {}^6H_{13/2}$ transition at 572 nm.

Bigotta et al. [48] reported laser operation in 0.5 at.% $Er^{3+}$-doped YAG polycrystalline ceramics developed by SPS starting from commercial precursor powder size of 271 nm. After obtaining SPS samples, they were treated with hot isostatic pressing (HIP), which helps in obtaining highly transparent ceramics and removing porosity. SPS was developed at 1450 °C for 2 h. After sintering, the sample was exposed to HIP under Ar atmosphere at 1400 °C, 190 MPa pressure for 15 h. Direct transmission was 75.8 and 82.7% at 400 nm and 1100 nm wavelength, respectively. Laser action with the slope efficiency of ~31% and optical efficiency of 20% was recorded.

Recently, Avital Wagner et al. [49] prepared $Nd^{3+}$ (i.e., 0.5, 1, 2, 3, and 5 at.%): YAG transparent ceramics by conventional SPS and compared with high-pressure SPS (HPSPS) method also. SPS process parameters were 1400 °C, 60 MPa, and 2 h time, while HPSPS parameters were 1300 °C, 300 MPa, and 1.5 h. The transparency of HPSPS-sintered samples was lower than those of conventional SPS-sintered samples due to the difference in the average pore size. The maximum emission intensity increased sharply up to 2 at.% $Nd^{3+}$ and then gradually decreased with increasing $Nd^{3+}$ concentrations for both sintering processes.

Spinel $MgAl_2O_4$ transparent ceramics—undoped and doped with rare-earth ions (0.1%-$Tb^{3+}$ and 0.1%-$Dy^{3+}$ ions)—were synthesized by SPS process at a sintering temperature of 1400 °C, the pressure of 72 MPa, and 10 min time. XRD patterns showed the absence of shifts in peaks position and absence of additional peaks in the sample, confirming that the dopant ($Tb^{3+}$ and 0.1%-$Dy^{3+}$ ions) completely incorporated into the polycrystalline structure. The absorption edge for undoped and doped spinel were 250 and 270 nm, respectively. An increase was observed in the band intensity located at 3.63 eV from pulse cathodoluminescence (PCL) spectra on the addition of dopant owing to the formation of intrinsic host defects [50].

## 5. Transparent Glass-Ceramics

This review focuses on the processing of glass-ceramics by SPS as a new approach that may improve the conventional heat-treatment process. In particular, it is a promising technique to obtain transparent GC materials. GCs are classified according to the presence of glass network formers within the composition, such as silicate, chalcogenide, and oxyfluoride GCs. Several aspects regarding GCs are under research, such as the influence of the size of the precursor particles on the kinetics of solid-phase SPS, why fine-grained crystalline phase within glassy matrix sinter faster when SPS is used, or the reasons behind dependence of density with sintering temperature.

*5.1. Silicate GCs*

The first report on glass-ceramics prepared by SPS was published by Riello et al. in 2006 [51], dealing with glass-ceramics in the $Li_2O–Al_2O_3–SiO_2$ (LAS) system doped with erbium ions. Small concentrations of $ZrO_2$, about 1.02–1.04 mol%, were added as nucleating agent and to promote mechanical strength.

The initial bulk glasses were prepared by melt quenching and further heat-treated at 1000 °C for crystallization to obtain bulk glass-ceramics.

Glass powders were also obtained by the sol-gel method and used for SPS sintering. Rapid pulses of 3 ms in length with the pattern of 12:2 = ON:OFF, pressure ranging between 35 and 53 MPa, temperature between 840 and 900 °C (200 °C/min), and holding time of 2–5 min were used as sintering parameters for SPS. Samples were heated at 850 °C, 35 MPa, 2 min; 840 °C, 53 MPa, 5 min; and 900 °C, 53 MPa, 5 min and designated as SPS1, SPS2, and SPS3, respectively.

The XRD patterns showed that SPS1 and SPS2 were amorphous samples, and SPS3 was crystalline (β-Eucryptite/β-quartz s.s), respectively, after SPS measurements. Only one sample (SPS3-900 °C/53 MPa/5 min) presented a crystalline fraction of 45% with the size of the crystals of 10 nm. It is interesting to point out that the SPS3 (900 °C/53 MPa/5 min) sample exhibited a density of 2.48 $g/cm^3$ very close to that obtained by melt-quenching (2.50 $g/cm^3$). The average Vickers mechanical microhardness values were 6.76 and 7.10 GPa for SPS3 and conventional glass-ceramic samples, respectively. The mechanical hardness was also lower for the SPS-sintered GCs sample compared to the melted one. As prepared, all the samples were translucent owing to porosity (less than 100 ppm), as shown in Figure 3a. The maximum size of porosity was found to be 5 μm, leading to light scattering and reduced transparency.

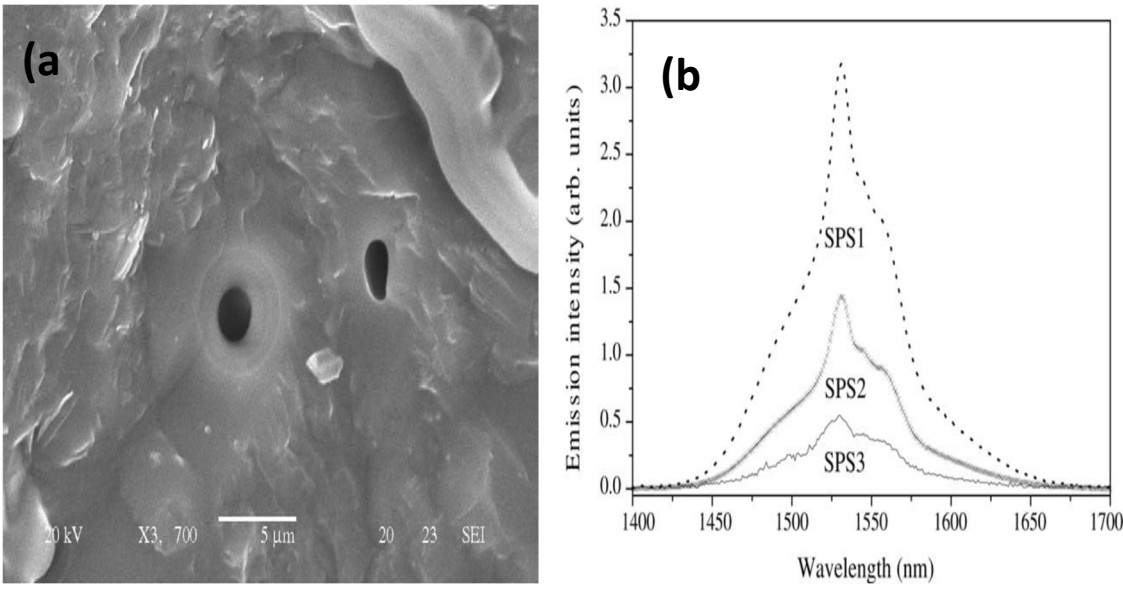

**Figure 3.** (**a**) SEM micrograph of SPS3, showing the residual porosity and (**b**) IR emission spectra of LAS-Er-Zr samples. Reprinted with permission from the Journal of the European Ceramic Society [51]. SPS, spark plasma sintering.

Initially, near infrared NIR emission spectra were measured for all the SPS samples and compared. All the samples showed an emission wavelength of 1530 nm corresponding to the $^4I_{13/2} \rightarrow {}^4I_{15/2}$ transition. The emission intensity decreased with the development of crystalline phases (intensity for SPS3 was about six times lower than for SPS1) because of higher erbium concentration (30 wt.%). Er ions relocated within the $ZrO_2$ phase, and its concentration became high enough to produce concentration quenching on luminescence intensity. Due to higher $Er^{3+}$ ion concentration,

agglomeration predominated, resulting in lower NIR emission intensity for the sample SPS3. SPS1 sample sintered at 850 °C/35 MPa/2 min conditions showed the highest NIR luminescence profile. NIR emission intensity of the SPS1-sintered sample was compared with bulk glass and glass-ceramic samples (Figure 3b). As observed, the near-infrared (NIR) fluorescence intensity was higher in bulk glass prepared by melt quenching with respect to the other two samples, such as bulk glass-ceramic sample and SPS1. The authors pointed out that porosity must be optimized for use in optical applications. However, the effects of glass powder particle size, preheat treatment of glass powder, pressure, temperature, and holding time were not studied.

Chen et al. in 2012 [52] densified for the first time a sol-gel-synthesized 45S5 Bioglass®-ceramics by SPS. They found non-transparent GC materials, but the goal was to show the suitability of a method of preparation followed for SPS, following a new way of the sol-gel process rather than the melt quenching method. The sol-gel material was obtained by mixing appropriate precursors. The dried 45S5 Bioglass® powder was pelletized under uniaxial pressure of 50 kPa. The pellets were subjected to conventional sintering at 700, 800, 900, 950, 1000, 1100, and 1200 °C and SPS sintering at 600, 750, 800, and 950 °C. Densities through conventional and SPS sintering were 89% and 92%, achieved at 1200 and 950 °C, respectively. High densification was obtained at much lower temperatures by the SPS process. No significant changes in density were observed at higher temperatures. Fine and homogenously distributed crystalline particles of the $Na_2Ca_2Si_3O_9$ phase were present in the glass matrix. The SPS-sintered samples presented fewer microvoids and a higher homogeneity together with fine crystalline particles, which provided enhanced mechanical characteristics. A Young's modulus (~110 GPa) and compressive strength ~110 MPa were achieved higher than those obtained by conventional heating.

López-Esteban et al. in 2014 [53] prepared soda-lime GCs for the dental application by SPS. The powders were sintered in vacuum at 750 °C for 3 min and a pressure of 32 MPa. XRD patterns revealed the formation of two crystalline phases: nepheline $(Na(AlSiO_4))$ and combeite $(Na_4Ca_4(Si_6O_{18}))$ with the crystalline size of below 5 μm. Density was 2.7 g/cm$^3$ without any porosity. Similarly, they followed cold isostatic pressure (CIP) to obtain soda-lime GCs and compared them with SPS-processed ones. In the CIP experiments, the samples were sintered at 750 °C, a pressure of 300 MPa for a time of 1 h. Density was 2.3 g/cm$^3$, with 17% of the final porosity size of below 100 μm. The surface roughness (Sa) was found to be low (0.7 μm) for the SPS-sintered samples and high (3.2 μm) for CIP samples.

Fatima Al Mansour et al. in 2015 [43] discussed and studied the effects of SPS on the microstructure of lithium disilicate (LDS) glass-ceramics and compared with conventional sintering. They compared and analyzed the effects of SPS on two materials—one was IPS e.max CAD, and another one was IPS e.max Press. IPS e.max CAD glass-ceramic samples were processed by spark plasma sintering (SPS) and conventional sintering (CS) for comparison. Samples were sintered at varying temperatures, heating rates, and pressures to analyze their significance on materials. It was noticed that an additional graphite phase was identified in XRD for the SPS-sintered sample owing to contamination from the graphite punch and die used in the SPS set up. To avoid this problem, the authors suggested using molybdenum foils and the possibility of reduction of surface contamination. However, there was a substantial increase in median crystal size between the conventionally-sintered at 840 °C and the SPS-sintered at 840 °C. They also noticed that applying higher temperatures was more effective for promoting grain boundary diffusion processes and crystallite growth, also for the IPS e.max CAD glass-ceramic samples. The IPS e.max CAD had a lower crystal fraction of the LDS phase (40%) compared to the IPS e.max Press (70%) after the heat treatment at 840–850 °C.

Le Fua et al. in 2017 [54] investigated on $xZrO_2$-(100-x) $SiO_2$ (x = 45 mol%, 55 mol%, 65 mol%) system, prepared by sol-gel method. Sieved precursor glass powders with 50–100 μm were further used for SPS methods. The XRD profiles of samples sintered under 1150 °C/60 MPa/5 min revealed that the only crystal phase in all the samples was tetragonal $ZrO_2$. The average nano-sized $ZrO_2$ crystal of the 45 Zr, 55 Zr, and 65 Zr samples was 29.5 nm, 35.1 nm, and 47.5 nm, respectively. The relative density of the 45 Zr, 55 Zr, and 65 Zr glass-ceramics was 3.82 g/cm$^3$, 4.14 g/cm$^3$, and 4.53 g/cm$^3$, respectively.

The crystal size of $ZrO_2$ and the density of samples increased with the concentration of $ZrO_2$. The authors confirmed that the detrimental effect of low-temperature degradation (LTD) of tetragonal $ZrO_2$ limited the usage as dental materials. TEM images revealed that nano GCs had much less grain boundary, and the silica encapsulating surface layer on nano-sized tetragonal $ZrO_2$ contributed to the high hydrothermal stability of nano GCs, which was useful for medical applications. The SPS-sintered nano GC had translucency and lower transmittance (20%) under direct transmission measurements. The text under the different samples was observed with the naked eye and was clearly recognized, when the blurry text appeared for 45 Zr, 55 Zr, and 65 Zr samples, respectively.

Sunil Kim et al. in 2017 [55] studied and prepared phosphor in glass (PIG) using the SPS process (SPS PIG) to be compared in terms of microstructure and optical properties with those of the PIG sintered in an electric furnace. The $20SiO_2$-$30B_2O_3$-$45ZnO$-$5Li_2O$ (mol%) glass frit was prepared using an electric furnace at 1200 °C for 30 min in air. This frit was milled and screened using a 100-μm-size mesh. Two types of phosphors—yellow phosphor (Y phosphor, $Y_3Al_5O_{12}$:$Ce^{3+}$) and red phosphor (R phosphor, $Ca_{0.2}AlSiSr_{0.8}N_3$:$Eu^{2+}$)—were selected and mixed with 5 vol% of the phosphor to convert glass frit into the PIG. These powder samples were SPS-sintered at 520 °C/40 MPa/10 min conditions at different heating rates and also sintered in an electric furnace at 630 °C for 30 min at a heating rate of 10 °C/min in air. The SPS sintered with yellow phosphor PIGs showed a smaller mean pore size and lower porosity than the glass frit plate, and the same composition sintered in an electric furnace. Similarly, the SPS-sintered red PIGs showed a lower porosity and smaller pore size than the glass frit plate and those sintered in an electric furnace. The pore properties were influenced more by the heating rate than by temperature. The porosity of the SPS red PIGs was slightly higher than that of the SPS-sintered yellow PIGs owing to the increased viscosity that prevents the densification based on the mass change and the shape of the phosphor. The SPS-sintered yellow PIGs showed a higher transmittance than the glass without phosphor and those sintered in an electric furnace of yellow PIG. In the case of the red PIGs, the SPS-sintered red PIGs showed a lower transmittance than the glass plate because of the influence of the R phosphors with a high refractive index, but a higher transmittance than red phosphor sintered in an electric furnace. As the porosity of the SPS PIGs decreased, the transmittance increased. Authors remarked that the presence of an appropriate number of small pores with diameters less than 5 μm in the PIG increased the phosphor intensity and luminous efficiency by increasing the scattering angle of the light that caused the increase in the interaction with the phosphor as compared to the PIGs with no pores.

## 5.2. Chalcogenide GCs

Delaizir et al. in 2010 [56] studied the $62.5GeS_2$–$12.5Sb_2S_3$–$25CsCl$ (mol%) composition approached by SPS in order to reduce the synthesis times compared with conventional melting and heat treatment processes. The glass was prepared by melt quenching, milled, and the prepared powder was introduced into the SPS system. SPS processing was conducted in a vacuum atmosphere, a pressure of 100 MPa for all samples, and varying holding times ranging from 2 to 120 min. The transmission spectra confirmed that with increasing holding time, bands were shifted toward longer wavelengths because of MIE scattering, and the absorption bands at 2.9, 4, 6.3, and 7.8 mm due to O–H, S–H, $H_2O$, and Ge–O bonding, respectively, were observed. The visible aspects of the as-prepared samples and corresponding XRD spectra are shown in Figure 4. As observed, the density of the SPS-sintered glass-ceramics increased with sintering time from 3.18 $g/cm^3$ (10 min) to 3.23 $g/cm^3$ (90 min). XRD and SEM analysis showed an increase of crystal size < 100 nm with increasing temperature, attributed to the combined effect of pressure applied during SPS and holding time. The intensity of the XRD peaks increased when holding time increased, meaning that the number of nuclei increased, and a fourth peak appeared at 31° related to CsCl crystals.

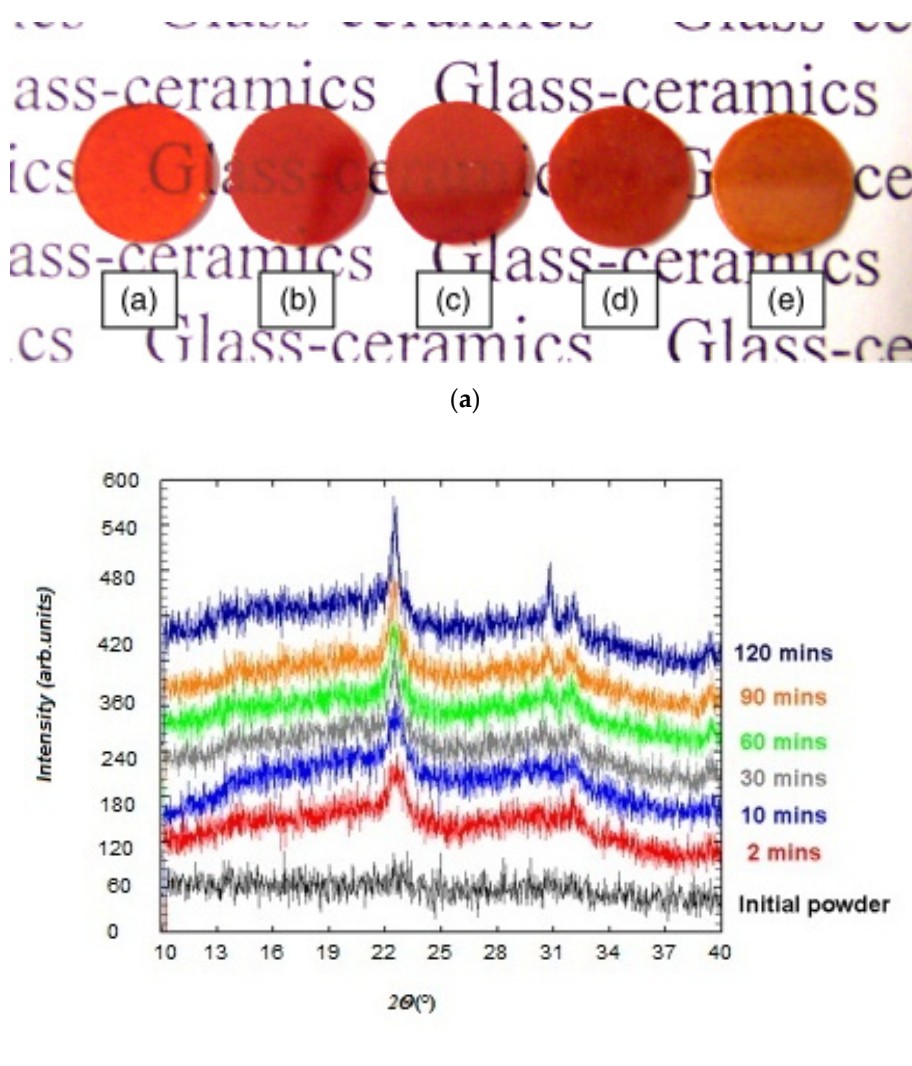

**Figure 4.** (**a**) Visual aspect of glass-ceramics samples obtained from the 62.5GeS$_2$–12.5Sb$_2$S$_3$–25CsCl-based glass powder for, respectively, spark plasma sintering treatment times of (a) 10 min, (b) 30 min, (c) 60 min,(d) 90 min, and (e) 120 min at 2901 °C under 100 MPa and (**b**) corresponding XRD patterns. Reprinted with permission from the Journal of the American Ceramic Society [56].

Mathieu Hubert et al. in 2011 [57] approached a different way for the preparation of glass-ceramics from chalcogenide amorphous powders of composition 80GeSe$_2$–20Ga$_2$Se$_3$. The glass powders were prepared by mechanical milling with 0 to 80 h and changed their color from gray to red. This was due to a decrease in particle size with an increase in mechanical milling and the chemical reaction that takes place between the raw materials. They introduced 80 h milled glass powders into a carbon mold (diameters from 8 mm to 36 mm) with a graphite/tantalum foil in the inner part of the mold as a carbon diffusion barrier. This powder was sintered under vacuum at 390 °C, slightly higher than the glass transition temperature (T$_g$), by SPS (pressure ~50 MPa) at different well times from 2 to 60 min.

The final density was 4.39 g/cm$^3$ for sintering at 390 °C under 50 MPa and 2 min of dwell time. The glass prepared by melt quenching showed an equivalent density value. The transmission spectrum of the obtained bulk glassy samples, whatever their dimension was, good in the mid-infrared range as observable in the picture taken by a thermal camera working in the third atmospheric window from 8 μm to 12 μm. Those GCs were opaque in the visible region. The incorporation of carbon from graphite mold was observed and confirmed by XRD analysis. They suggested that this contamination could be avoided by pre-compacting the powder (making pellet) at room temperature before introducing in the

SPS process. For avoiding contamination, in the next experiment, tantalum foil was used as a diffusion barrier. XRD patterns showed no evidence of either carbon or tantalum.

In order to promote crystallization inside the glass matrix, they played with dwelling time (2–60 min) of the SPS experiment. The XRD obtained for 30 min time clearly showed that the $Ga_2Se_3$ (or $Ga_4GeSe_8$) crystalline phase was nucleated, and a good homogeneity inside the glassy matrix was elucidated. For the treatment time (60 min), at the same temperature and pressure, a second phase ($GeSe_2$) was formed, and no cracks were observed. Transmittance at lower dwelling time (2 min) was ~55%, and at longer dwelling time was ~20% due to the MIE scattering effect of the precipitated nanoparticles. The base glass composition had ~70% transmittance. The Vickers hardness was 201, 203, and 167 Hv at a dwelling time of 2, 30, and 60 min, respectively. The Vickers hardness followed a similar trend with a rise in dwelling time up to 30 min time.

In 2012, Delaizir et al. [58] examined the sintering mechanisms in $80GeSe_2–20Ga_2Se_3$ chalcogenide glass composition, as well as crystallization precipitation by SPS. The sintering of glass occurred by viscous flow, reducing surface energy of a porous compact through neck growth and densification involving the deformation of initial particles. Different mechanisms leading to the production of glass-ceramics had been hypothesized. The first one would be the densification of glassy powder through viscous sintering, followed by subsequent devitrification of the matrix. The second mechanism would imply densification and concurrent gradual crystallization of the matrix through the growth of the neck between glassy particles. For one given glass composition, the corresponding crystalline phase had a considerably higher viscosity than the amorphous phase, so the sintering of polycrystalline materials was more difficult than the amorphous phase. This suggested that the first mechanism described above, i.e., achieving full density prior to any significant crystallization, should be favored.

To confirm the involved mechanism, the influence of the dwell temperature (250–390 °C), where the full densification of glass was achieved without crystallization, on the different dwell timings (2, 15, and 60 min) was studied, keeping the pressure constant (50 MPa). At 300 °C, the shrinkage started intensively. The viscous flow between particles led to the formation of necks until 350 °C and the plastic deformation of particles. For increasing temperatures, from 250 °C to 390 °C, and 2 min dwell time, the particles started to soften and melt together, but a lot of intergranular porosity (27%) was observed. At 330 °C, viscous flow between particles led to the formation of necks, and, at 390 °C, the particles were totally melted together, and no porosity was observed. The increase in temperature led to an increase in compactness and residual porosity diminution. Similarly, the compactness increased and induced partial devitrification of $Ga_2Se_3$ (or $GaGe_4Se_8$) when the dwell time increased. The authors explained that the densification mechanism occurred prior to devitrification.

Anthony Bertrand et al. in 2013 [59] prepared transparent tellurite glasses and glass-ceramics in the $85TeO_2–15WO_3$ (mol%.) system by SPS with the main emphasis on understanding and eliminating carbon contamination by studying the effect of initial amorphous particles size, the effect of different carbon diffusion barriers (alumina, tantalum foil, and platinum), as well as the effect of a pressureless sintering step prior to SPS was investigated. They used three sizes of precursor particles, such as coarse, fine, and bulk particles for SPS sintering.

SPS measurements at a constant pressure of 50 MPa were sintered. The carbon contamination was higher in the case of the fine particles, leading to dark glass bulks. The use of a diffusion barrier, such as platinum, tantalum foil, or alumina, limited carbon contamination. Pressureless pellets heat-treated at $T_g$ +30 °C for 1 h prior to SPS also limited carbon contamination. Yellow glasses with relatively high optical transparency properties were obtained by SPS. By increasing the dwell time at constant pressure and temperature during the SPS experiments, the non-centrosymmetric δ-$TeO_2$ phase crystallized, which generated a second harmonic signal. The increase in dwell time led to an increment in crystalline fraction, low porosity, and higher density. The optical transmission spectra obtained by SPS was compared with the glass sample prepared by the conventional melt-quenching MQ technique. A lower optical transmission at short wavelengths was noticed for an SPS glass sample due to light scattering by residual porosity, small inclusions/crystals, or residual pollution. However, despite this reduction,

the optical transmission was high (up to 70% transmission) for these glasses prepared by SPS. From the graphite mold, the oxygen, water, or carbon dispersion caused impurities on the surface of powders. The surfaces of the sample seemed to be rough and confirmed the formation of the crystalline phase.

Xue and co-workers, in 2013 [60], studied $GeS_2$ glass-ceramics for infrared applications. $GeS_2$ glasses were obtained by melt quenching and then mechanically ground in the air or in a glove box. Two different ways were used to prepare $GeS_2/\beta$-$GeS_2$ glass-ceramics. The first way, glass of $GeS_2$ heated at 490 °C for some hours to obtain the GC sample. The second way, initially, $\beta$-$GeS_2$ crystals were synthesized by controlled crystallization of $GeS_2$ glass powders at 493 °C for 250 h. $\beta$-$GeS_2$ crystals were mixed thoroughly in a planetary grinder with $GeS_2$ glass powder (85 mol%.) for about 3 h to obtain a homogeneous mixture. These powders were further sintered by SPS. Two samples were prepared, namely, GC1 and GC2. These samples were prepared under the same environmental conditions (450 °C and 50 MPa) with varying dwell time, i.e., 10 min (GC1) and 15 min (GC2). GC1 was processed under the glove box and GC2 under the air atmosphere.

At the dwell timing of 10 min, small crystal fraction (15 mol%) of $\beta$-$GeS_2$ crystal was formed. Further, with the increase of dwell time (GC2), crystal growth was increased, and the crystal looked bigger in size. Some pores at the boundary between glass and crystals led to a lower density (GC2). Higher porosity was observed in GC2 samples compared with the GC1 samples. High porosity derived from the existence of oxide through the air atmosphere on the powder surface. GC1 showed higher transmission (from IR) in comparison with GC2 due to strong scattering caused by pores at the boundary of glass-crystal. The authors proposed that these scattering effects were minimized by the full crystallization of $\beta$-$GeS_2$ of smaller size in base glass. From the transmission curves, the absorption bands at 2.8 μm and 4 μm, wavelengths related to the hydroxyl group, were noticed. It was worthy to note that grinding in the glove box was a more efficient approach than air to decrease the contamination by water. Nevertheless, the high transmission window at 10.5 μm was only about 30%. Strong scattering was mainly due to pores at the boundary glass/crystal. This scattering, still affecting the lower wavelengths, were minimized in the GC1 sample as crystals were homogeneously dispersed.

Cui et al. in 2015 [61] prepared tellurium-based GCs by SPS. $(Te_{85}Se_{15})_{60-0.6x}As_{40^-0.4x}Cu_x$ (x = 0, 10, 16.7, 20, 25) glasses were labeled as TEA1, TEA2, TEA3, TEA4, and TEA5, respectively. These tellurites were prepared using the melt quenching method and also the $Bi_{0.5}Sb_{1.5}Te_3$ (BST) ceramic system. The powder size (<50 μm) of both tellurite glass and the BST powders were ball milled and sieved. These powders were sintered at 463 K for 10 min of holding time under a pressure of 40 MPa. Different proportions (—0, 10, 30, and 50%) of BST powders were mixed with tellurite glass powders and named as BST0 (99%), BST10 (99.4%), BST30 (98.1%), and BST50 (97.7%), respectively. During synthesis and due to the semiconductor nature of both the glass and BST ceramic system, Joule heating occurred in both the carbon die and the powder, generating densification through the viscous sintering of the glass. Thus, the sintering temperature of glass-ceramics (463 K) was much lower compared with pure BST (>700 K).

### 5.3. OxyfluorideGCs

For the last ten years, the GlaSS group at Ceramics and Glass Institute (CSIC) focused attention on the preparation of transparent glass-ceramics with low-phonon fluoride crystals, such as $LaF_3$, $NaGdF_4$, $NaLuF_4$, $KLaF_4$, etc., with rare-earth ions as dopants for optical applications [29,30,62–78]. The main emphasis was on maximizing the crystalline fraction, maximum incorporation of the RE dopant, and further enhancement in luminescence profile. More efforts were developed towards understanding the mechanisms of nucleation and crystallization of fluoride nanocrystals. All these materials were prepared by homogeneous nucleation by conventional melt quenching and then heat treatment processes at temperatures slightly higher than $T_g$.

$LaF_3$ crystals smaller than 20 nm diameter were precipitated from the base glass and formed within phase-separated droplets. The glass transformed into nano-glass-ceramics by heating at 645 °C for 20 h [61]. Oxyfluoride GCs doped with 0.1 and 0.5 Pr and co-doped with 0.1–0.5 Pr–Yb and

0.5–1 Pr–Yb GCs treated at 620 and 660 °C were perfectly transparent, owing to the formation of nanosized crystals of $LaF_3$. With the increasing concentration of dopants, the nuclei density increased, but the nuclei size became smaller [63]. Similarly, other dopants were studied to obtain $LaF_3$-based glass-ceramics, using thermal treatments in the range 1–5 h at 620 °C [64,65].

Transparent oxyfluoride GCs containing $NaGdF_4$ nanocrystals doped with 0.1 $Pr^{3+}$ and 0.5 $Pr^{3+}$ and co-doped with 0.5 $Pr^{3+}$-2 $Yb^{3+}$ ions (mol%) were also obtained [66]. The X-ray diffraction (XRD) and high-resolution transmission electron microscopy (HRTEM) confirmed the precipitation of $NaGdF_4$. The kinetics at 550 °C as a function of the dwelling time resulted in an increase of the nanocrystal size up to 12.5 nm and remained constant after 80 h. In addition, XRD of doped and co-doped GC heat-treated at 550 °C during 80 h showed that the nanocrystal size increased from 13 up to 30 nm with the dopant concentration.

Transparent $NaLuF_4$ glass-ceramics doped with 0.5 mol% $Er^{3+}$ and 0.5 $Er^{3+}$ -x$Yb^{3+}$ (x = 2, 4 mol%) were prepared by the melting-quenching method, and, after suitable thermal treatment at 600 °C for 20 h, cubic $NaLuF_4$ nanocrystals ranging from 10 to 50 nm (depending on dopant concentration) were obtained. Up-conversion green emission related to $^2H_{11/2}$, $^4S_{3/2} \rightarrow ^4I_{15/2}$ transition, and red emission related to $^4F_{9/2} \rightarrow ^4I_{15/2}$ transition were observed in all GCs [67].

Cubic (α-phase) and hexagonal (β-phase) $KLaF_4$ glass-ceramics were also obtained doped with four different $Nd^{3+}$ concentrations (0.1–2 mol%). These samples were prepared by heat treatment at 590 °C from 1 to 150 h and at 660 °C from 1 to 192 h. The phase evaluation depended on the $Nd^{3+}$ dopant concentration. In the case of 0.1 mol% of the $Nd^{3+}$-doped sample, only after heat treatment at 660 °C for times longer than 144 h, β-$KLaF_4$ nanocrystals were formed. In the case of 0.5 mol% of $Nd^{3+}$-doped sample, β -$KLaF_4$ nanocrystals were formed only for times longer than 15 h, and the authors concluded that hexagonal nanocrystals needed longer growth times [68].

On the other hand, the sol-gel method was also explored. Sol-gel oxyfluoride $LaF_3$ glass-ceramics were obtained with very fast treatments (1 min) at 550 °C temperature. It was proposed that the crystallization process of $LaF_3$ in sol-gel materials was not diffusion-controlled nucleation and growing process, but a chemical reaction followed by the fast precipitation of crystals [76]. Similarly, undoped and doped 0.5 $Eu^{3+}$ (mol%) $GdF_3$ transparent GCs were prepared by applied heat treatment at 550 °C temperature for 1 min to 8 h. The authors noticed that methyl triethoxysilane (MTES)-silicon precursor, in addition to tetraethyl orthosilicate (TEOS), improved the mechanical properties. Dopant $Eu^{3+}$ ion incorporated successfully in the crystals, and XRD peaks of the hexagonal and orthorhombic phases of $GdF_3$ were shifted to larger 2θ angles. By careful observation of luminescence data, efficient energy transfer from $Gd^{3+}$ to $Eu^{3+}$ in the nanocrystals was noticed, as well as an increase in the $Eu^{3+}$ reddish visible emission [77]. Different composition of doped $90SiO_2$-$10NaGdF_4$ GCs was also investigated with a similar approach. These GCs were fully transparent and homogeneous when heating the xerogels at 550 and 600 °C from 1 min to 8 h. Samples treated for >8 h appeared opaque due to larger NCs size or due to the formation of clusters. The structural analysis—XRD and HRTEM—confirmed the precipitation of $NaGdF_4$NCs in two crystalline phases (cubic and/or hexagonal) with a size ranging between 4 and 24 nm, depending on the Na:Gd ratio, heating temperature, and time [78].

This previous extensive work on transparent oxyfluoride GCs by conventional heat treatment was a time-consuming process (from 1 to 192 h), thus constituting its main disadvantage. We realized that it is possible to reduce this time using the SPS technique. Additionally, other properties like full densification, the increment of crystalline fraction, and lowering sintering temperature could also be addressed. Moreover, in the case of sol-gel powders, SPS opened the possibility of getting highly dense and mechanically stable materials that could not be obtained in bulk by the conventional route.

Recently, the GlaSS group has been focused on SPS sintering of rare-earth-doped transparent oxyfluoride GCs looking for faster densification in a short sintering time than conventional heat-treatment process. These oxyfluoride GCs have various potential applications, such as host materials for solid-state lasers and optical materials due to low phonon energy of fluoride crystal within the glassy matrix. In order to achieve optical transparency, the platinum foil was used to cover

the graphite die mold in order to decrease or even eliminate carbon contamination. The obtained samples are shown in Figure 5. This example corresponded to Nd-doped KLaF$_4$ glass-ceramics. As can be seen from Figure 5, it is clear that platinum was effective in order to avoid carbon contamination coming from the graphite die.

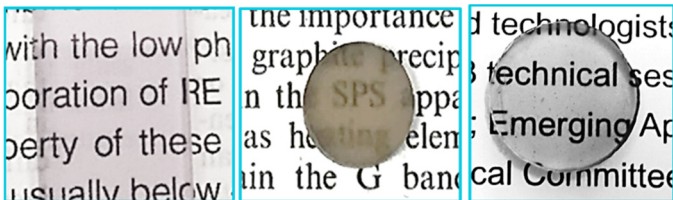

**Figure 5.** Rare earth-doped transparent oxyfluoride GCs prepared by (from left to right) melt quenching, SPS with uncovered graphite die, SPS with platinum foil.

Once reducing the carbon contamination, KLaF$_4$ transparent GCs were obtained. During the SPS process, the loading pressure was 22 MPa, sintering temperature was 700 °C, with different holding times (10–20 min) and different particle sizes (>63 μm and 63–100 μm). The densified samples had a cylindrical shape, with a diameter of 15 mm and a thickness of about 3 mm. The samples were polished on both sides for optical measurements. Among all GC samples, those doped with 0.5 Nd, a particle size 63–100 μm, and a treatment time of 20 min had the highest transparency. The increase of the holding time favored the pores' elimination.

Crystalline phases were investigated by using X-ray and are shown in Figure 6a. The diffraction peaks were associated with the crystallization of the cubic (α) and hexagonal (β) polymorph of KLaF$_4$. The crystal size was nearly between 19–23 nm for cubic phase KLaF$_4$ and 9–12 nm for hexagonal phase KLaF$_4$. This size of the KLaF$_4$ crystalline phase was also confirmed by transmission electron microscopy (TEM), as shown in Figure 6b.

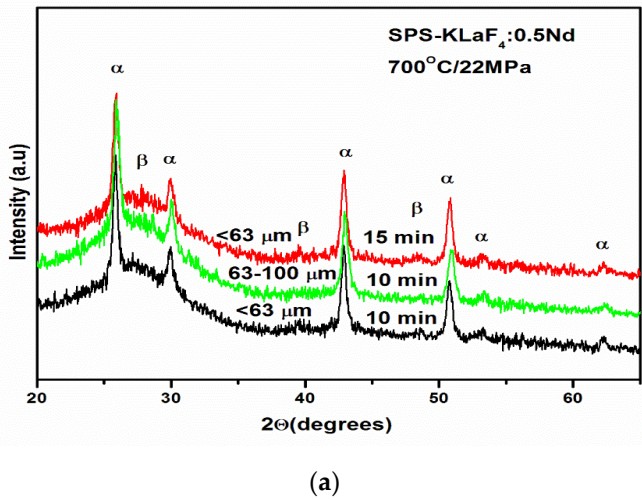

(**a**)

**Figure 6.** *Cont.*

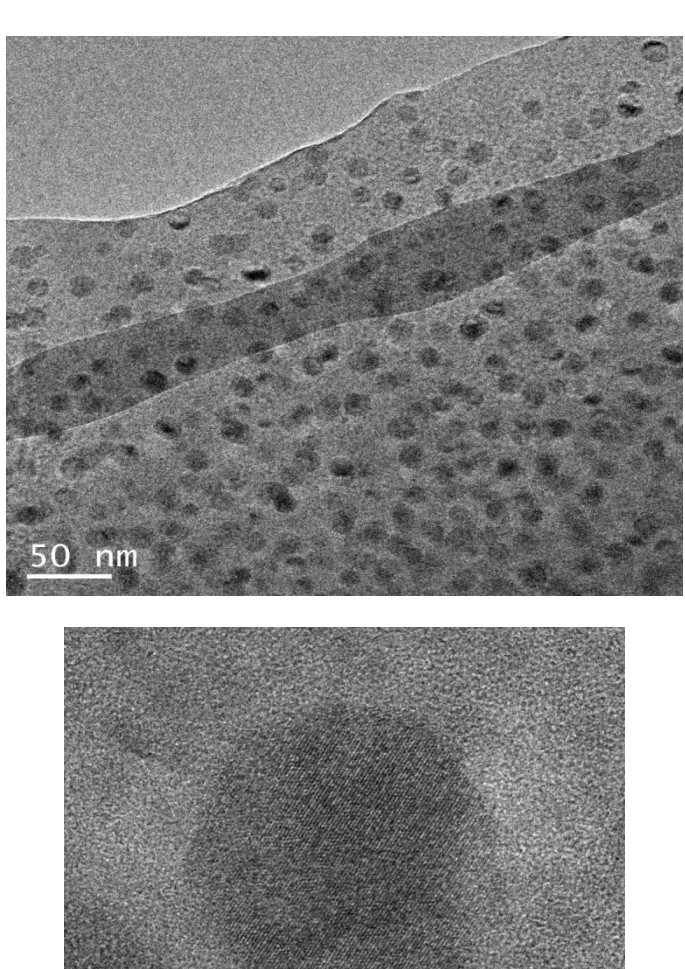

(**b**)

**Figure 6.** (**a**) XRD patterns for KLaF$_4$: 0.5 Nd$^{3+}$-doped transparent glass-ceramics (GCs) with different holding time and particle sizes, (**b**) TEM images of 0.5 Nd$^{3+}$-doped transparent GCs.

Oxyfluoride GC samples have high density and are transparent. To the best of our knowledge first report on oxyfluoride-based highly transparent GCs was obtained by SPS sintering. All the technical and experimental details could be found in [79]. In addition to the above GC materials, other crystalline transparent GC phases, such as NaLaF$_4$ and NaLuF$_4$, have been also successfully prepared and characterized and would be reported by the group soon. We have concluded that this method is suitable for the preparation of transparent GCs for optical applications. Table 2 summarizes the main differences found between the SPS sintering and conventional method of melting, followed by thermal treatment in terms of processing and properties.

**Table 2.** Comparison of characteristics on spark plasma sintering (SPS) vs. conventional method for the preparation of transparent glass-ceramics.

| Process/Properties | SPS Sintering | Conventional Method |
| --- | --- | --- |
| Temperature rise rate | Excellent | Difficult |
| Temperature cooling rate | Fast | Slow |
| Homogeneity | Excellent | Fair |
| Temperature rise time | Fast | Slow |
| Holding time | Short | Long |
| Influence of pressure | Excellent | Absence |
| Transparency | Excellent | Good |
| Grain boundary controlled sintering | Excellent | - |
| Control of the nanometric size of the crystals | Excellent | Excellent |
| Carbon contamination | High | None |
| Densification | Excellent | Excellent |
| Mechanical strength | Excellent | Good |
| Processed using | Powder particles | Bulk sample |
| Mass Production | Difficult | Excellent |

## 6. Conclusions and Future Prospects

This paper review focuses on the importance of understanding new processing approaches to guide scientists and engineers in the development of alternative and superior materials. Most optical devices are equipped with single crystals, ceramics, glasses, or glass-ceramics owing to their high transmittance. In particular, transparent glass-ceramics have high demand and applications in various fields. Transparent GCs have been produced by melt quenching or sol-gel method, followed by heat treatments in order to precipitate the desired phases and with control of crystalline growth within the glass matrix. The use of an alternative processing method, such as SPS, provides various advantages. It is a time-saving process, and final materials show high mechanical strength compared to the conventional procedure. Although many techniques, such as HP, HIP, and CIP, are available, there is inadequate knowledge of sintering in the development of functional GCs prepared by the SPS method. Many efforts have been made to understand the underlying principle of processing mechanism, problems and their solutions, and possible applications, which still present limitations. The final materials and their properties are influenced by the processing parameters, such as temperature, pressure, time, etc. Several papers have been reviewed to follow a better way to prepare and control crystal growth. Concerning the microstructure, normally a single crystalline phase is preferred rather than mixed or complex phases, grain growth is undesirable, and porosity must be close to zero, as well as carbon contamination. Carbon contamination can be avoided or suppressed by using tantalum/molybdenum/platinum foils as a carbon diffusion barrier and pre-compacting glass powder below the $T_g$ of the glass. An increase in the dwelling time normally improves the compactness of GCs via sparks created at the time of sintering and reduces porosity, favoring the growth of necks formed between the particles. Some authors suggest that applying higher temperatures up to certain limits enhance full densification. It is essential to investigate and to find out the dominant densification mechanisms. Transparency can be improved by maximizing the crystallization of nanocrystals in the base glass and thereby eliminating the scattering effects at the boundary between the glass and crystal phase.

In the future, it is essential the standardization of the sintering parameters to enable and approach towards easy GCs processing. These materials have great potential due to their optical functionality for photonic applications. Some properties, such as high mechanical strength, laser threshold damage capacity of GCs, and high luminescence efficiency, must be still optimized to promote SPS synthesis as an alternative route. In addition, future research directions regarding optical devices should also include the development of a better and deeper fundamental understanding of the processing-structure-optical property relationship of the materials as well as of the device functioning. The answer is crucial for a

deeper understanding and technology uptake on large mass-scale production of materials through SPS, which is the main drawback of the SPS system in industrial sector usage. Unraveling the mechanisms behind material processing conditions and crystallization is certainly of high interest for further tuning the material superior mechanical, optical, and transmittance properties. Growth of halogen (fluoride, chloride, bromide, and iodide)-based nanocrystalline phase, which has low phonon energy instead of oxide-based materials and high transmission properties, has been a valuable research issue for further development and investigation of optical materials due to their unique optical properties. These materials are going to broaden their impact on optical devices and optical components. We expect that this review paper could be helpful for upcoming researchers and industrialists who have difficulties in developing new luminescent materials by SPS sintering technique.

**Author Contributions:** Literature investigation, writing-original draft preparation, B.S.; project administration, funding acquisition, D.G.; supervision, project administration, writing—review and editing, A.D.; conceptualization, literature investigation, methodology, supervision, writing—review and editing, project administration, funding acquisition, M.J.P. All authors have read and agreed to the published version of the manuscript.

**Funding:** Funding from the project MAT2017-87035-C2-1-P/2-P (AEI/FEDER, UE) is acknowledged. This paper is a part of the dissemination activities of project FunGlass. This project has received funding from the European Union's Horizon 2020 research and innovation program under grant agreement No. 739566. Funding from the project VEGA 1/0527/18 is gratefully acknowledged.

**Conflicts of Interest:** The authors declare no conflicts of interest.

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
