# Peer review of "Glass-Ceramics Processed by Spark Plasma Sintering (SPS) for Optical Applications"

_applsci, doi:10.3390/app10082791_

Round 1

Reviewer 1 Report

Within the manuscript the authors have collected important articles for both transparent ceramics synthesized by SPS technique and also silicate and chalcogenide glass-cerami…the references are relatively old (around 10 years old). However, the manuscript is useful for the readers because it provides a full picture the advantages and the possibilities of the method in a proper manner.

It has to be noticed that the authors recently proposed the SPS method for the oxyfluoride glass-ceramic synthesis and there is also a section about

the new directions, new questions, new problems to be solved, new improvements to be made.

My only comment is related to the lack of figures…The text would be more clear by using visual information like figures…two are not enough for a review paper. Therefore I suggest to used nice figures (i.e. for the SPS mechanism) …and properties (TEM/SEM, transparency, luminescence,…) depending on the authors choise.

Author Response

Thank you very much for your suggestions. As per your comments, we revised manuscript accordingly and included more figures in the revised manuscript. Changes were indicated in red color for easy understanding.

Reviewer 2 Report

This paper is a review on the preparation process of glass-ceramics by SPS and the properties of the prepared glass-ceramics. Many researches on them are described in this review. So this review is very helpful for researchers and engineers who study and develop the glass-ceramics. However, the difference in the features of the preparation process of glass-ceramics by SPS and conventional methods and the difference in the properties of the glass-ceramics prepared by SPS and conventional methods are not clear.

If possible, please show their differences in Table.

In addition, there are many typing errors in this paper. Please correct them. 

Author Response

Thank you very much for your comments and suggestions to improve our manuscript. As per your comments, we revised manuscript accordingly and included a able to get a better idea in the Page No 27.

Also, corrected typing errors with help of native English experts.